# Machine Learning Algorithms for the Retrieval of Canopy Chlorophyll Content and Leaf Area Index of Crops Using the PROSAIL-D Model with the Adjusted Average Leaf Angle

Qi Sun [1,2,3,4], Quanjun Jiao [2,4,*], Xidong Chen [5], Huimin Xing [1,3], Wenjiang Huang [4,6] and Bing Zhang [4,6]

1 Henan Engineering Technology Research Center of Ecological Protection and Management of the Old Course of Yellow River, Shangqiu Normal University, Shangqiu 476000, China
2 International Research Center of Big Data for Sustainable Development Goals, Beijing 100094, China
3 Henan Agricultural Remote Sensing Big Data Development and Innovation Laboratory, Shangqiu Normal University, Shangqiu 476000, China
4 Key Laboratory of Digital Earth Science, Aerospace Information Research Institute, Chinese Academy of Sciences, Beijing 100094, China
5 College of Surveying and Geo-Informatics, North China University of Water Resources and Electric Power, Zhengzhou 450046, China
6 College of Resources and Environment, University of Chinese Academy of Sciences, Beijing 100049, China
* Correspondence: jiaoqj@radi.ac.cn; Tel.: +86-10-82178181

**Abstract:** The canopy chlorophyll content (CCC) and leaf area index (LAI) are both essential indicators for crop growth monitoring and yield estimation. The PROSAIL model, which couples the properties optique spectrales des feuilles (PROSPECT) and scattering by arbitrarily inclined leaves (SAIL) radiative transfer models, is commonly used for the quantitative retrieval of crop parameters; however, its homogeneous canopy assumption limits its accuracy, especially in the case of multiple crop categories. The adjusted average leaf angle ($ALA_{adj}$), which can be parameterized for a specific crop type, increases the applicability of the PROSAIL model for specific crop types with a non-uniform canopy and has the potential to enhance the performance of PROSAIL-coupled hybrid methods. In this study, the PROSAIL-D model was used to generate the $ALA_{adj}$ values of wheat, soybean, and maize crops based on ground-measured spectra, the LAI, and the leaf chlorophyll content (LCC). The results revealed $ALA_{adj}$ values of 62 degrees for wheat, 45 degrees for soybean, and 60 degrees for maize. Support vector regression (SVR), random forest regression (RFR), extremely randomized trees regression (ETR), the gradient boosting regression tree (GBRT), and stacking learning (STL) were applied to simulated data of the $ALA_{adj}$ in 50-band data to retrieve the CCC and LAI of the crops. The results demonstrated that the estimation accuracy of singular crop parameters, particularly the crop LAI, was greatly enhanced by the five machine learning methods on the basis of data simulated with the $ALA_{adj}$. Regarding the estimation results of mixed crops, the machine learning algorithms using $ALA_{adj}$ datasets resulted in estimations of CCC (RMSE: RFR = 51.1 µg cm$^{-2}$, ETR = 54.7 µg cm$^{-2}$, GBRT = 54.9 µg cm$^{-2}$, STL = 48.3 µg cm$^{-2}$) and LAI (RMSE: SVR = 0.91, RFR = 1.03, ETR = 1.05, GBRT = 1.05, STL = 0.97), that outperformed the estimations without using the $ALA_{adj}$ (namely CCC RMSE: RFR = 93.0 µg cm$^{-2}$, ETR = 60.1 µg cm$^{-2}$, GBRT = 60.0 µg cm$^{-2}$, STL = 68.5 µg cm$^{-2}$ and LAI RMSE: SVR = 2.10, RFR = 2.28, ETR = 1.67, GBRT = 1.66, STL = 1.51). Similar findings were obtained using the suggested method in conjunction with 19-band data, demonstrating the promising potential of this method to estimate the CCC and LAI of crops at the satellite scale.

**Keywords:** crop chlorophyll; crop LAI; PROSAIL; average leaf angle; machine learning

## 1. Introduction

Crops are the world's primary food source, and the maintenance of the global food supply is largely dependent on crop management and agricultural production. Real-time, effective, and accurate access to crop biophysical parameters is the main means of

monitoring crop growth to carry out accurate agricultural management. At the canopy scale, the leaf area index (LAI) determines the amount of canopy foliage, which is directly related to the aboveground biomass of the canopy [1]. The canopy chlorophyll content (CCC) characterizes the photosynthetic capacity of the crop community and is the product of leaf chlorophyll content (LCC) and LAI. In addition, CCC is a key indicator affecting solar-induced chlorophyll fluorescence (SIF) [2] and the maximum photosynthetic carboxylation rate ($V_{cmax}$) [3]. The accurate and timely estimates of these parameters in large-scale areas are crucial for crop productivity assessment, field irrigation and fertilizer administration, pest and disease control, and other forms of agricultural management. However, traditional measurements are destructive, inefficient, labor-intensive, and resource-intensive and are therefore unsuitable for large areas. Remote sensing, on the other hand, provides a non-destructive and efficient method for estimating the LAI and CCC.

Using remote sensing technology, retrieving CCC and LAI primarily involves analyzing vegetation growth based on the distinguishing characteristics of crop canopy reflectance curves in remote sensing images. The red-edge region is highly responsive to chlorophyll and is frequently employed as distinctive spectra for estimating chlorophyll content. Meanwhile, the sensitivity of LAI to spectral reflectance runs throughout the entire spectral range (400–2500 nm). The retrieval of the CCC and LAI from canopy reflectance spectra has been carried out using two different types of approaches, namely empirically based [4,5] and physically based [6,7]. Empirical statistical models rely on conducting correlation analyses between spectra or spectral combinations that are sensitive to vegetation parameters and then retrieving the vegetation parameters. Due to the convenience of vegetation indices, there has been a proliferation of such indices developed for retrieving vegetation parameters. For instance, the MERIS terrestrial chlorophyll index (MTCI) is frequently used for estimating chlorophyll content [8], while the normalized difference vegetation index (NDVI) is commonly used to estimate LAI [9]. Additionally, numerous studies have assessed the utility of predictive equations calibrated on PROSAIL-generated or in situ data and full spectrum methods [10,11]. Although the empirically based methodology using vegetation indices (VIs) directly links biophysical parameters and VIs, this approach is typically impacted by the canopy architecture and the soil background in large-scale regions [7,12]. The physically based strategies, on the other hand, have been found to be reliable and suitable for a variety of circumstances because of their reliable physical mechanisms and substantial data backing. Physically based approaches generally refer to radiative transfer models (RTMs), which are based on physical laws describing the spectral/orientation variation of canopy reflectance with canopy, leaf, and soil background features [11,13,14]. They have advanced quickly in recent years, with scattering by arbitrarily inclined leaves (SAIL) [15] being the most extensively used canopy RTM and properties optique spectrales des feuilles (PROSPECT) [16] being the most popular model for the optical characteristics of leaves. The most well-known canopy RTM, the PROSAIL model, which combines the PROSPECT and SAIL models, has been commonly employed to retrieve the biochemical and structural variables of crops [17–20]. The main method for obtaining the parameters from the sizable dataset produced by RTMs is the use of a lookup table (LUT). The LUT is scanned to identify the elements with the highest correlation to measured reflectance and associated parameters, by selecting entries that best match recorded spectra with empirical data. However, due to the sluggish and inefficient per-pixel entry-based iterative calls, LUT-based solutions are primarily constrained by their computing cost [17,21].

With the use of machine learning, complex nonlinear relationships can be fitted more effectively to estimate crop traits by training RTM-based simulations [22–24]. Numerical optimization techniques can aid in resolving inverse problems in remote sensing by identifying the optimal parameter values that minimize a cost function, which quantifies the discrepancy between the model predictions and the observed data [13,25–27]. To simplify RTM inversion, a commonly used approach is to employ predictive equations, which are empirical relationships between the surface parameters and spectral indices or transformations. These equations are established based on synthetic spectra generated by an RTM

and subsequently applied to actual remote sensing imagery for rapid estimation of surface parameters [10,28]. The hybrid approach, which combines machine learning and RTMs, introduces a powerful and promising new tool for quantitative vegetation index retrieval. In recent years, hybrid techniques have received a great deal of attention and have been used to retrieve vegetation information [29,30]. The support vector machine (SVM) performs well in high-dimensional spaces, yielding results for vegetation parameter retrieval that are comparatively robust [31]. Random forest regression (RFR) [32,33], extremely randomized trees (extra trees) regression (ETR) [34], and the gradient boosting regression tree (GBRT) [35,36] are typical ensemble learning models that have received extensive attention in vegetation parameter retrieval due to their strong learning capabilities and stable performance. By employing an existing algorithm and some type of strategy, ensemble learning entails the creation of a group of "individual learners" from training data. GBRT is a boosting algorithm that uses a serial approach to train individual learners, and it relies on dependencies between them [37]. The RFR and ETR models are components of the bagging algorithm, which can be taught concurrently with little to no significant interdependence between individual learners during training [38]. Despite this, they seek to combine poor learners to achieve noticeably improved generalization performance. In contrast, another approach called stacking involves the construction of a new model by using a number of powerful foundation models that have already been trained [39–41]. Although the SVM, RFR, ETR, and GBRT methods have all demonstrated excellent machine learning capabilities for the retrieval of vegetation parameters, the methods differ in their regression capabilities. Moreover, the robustness of the models generated by each depends greatly on the sample size and quality, the study area, and the data source [28]. The technique of stacking multiple machine models is currently being highlighted, as it is considered to increase the reliability of the classification or retrieval findings [42–45].

RTM-based crop canopy reflectance is complicated by a variety of vegetation variables, particularly the canopy structure, which compromises the effectiveness of both empirical and sophisticated machine learning algorithms [17,46,47]. In addition to leaf biochemical parameters, various structural elements such as the LAI, leaf inclination angle distribution (LAD), soil background, and leaf clumping at various scales also affect canopy reflectance [48–51]. Both empirical and physically based methods suffer from the ill positioning of the retrieval where contrasting canopy settings can lead to almost identical spectral signatures [52–54]. For the retrieval of chlorophyll and the LAI at the canopy scale, the LAD is an influential structural parameter that determines the canopy spectral characteristics [55]. The crop leaf distribution in the PROSAIL model is essentially defined by six average leaf angles (ALAs). In actuality, however, this assumption is at odds with the random distribution of crop leaves. The ALA input in the PROSAIL model has been modified to make the simulated spectrum match the measured spectrum of the crop [19,56]. Jiao et al. [19] used the random forest (RF) algorithm to obtain the adjusted ALA of wheat and soybean crops to estimate their CCC values by coupling the PROSAIL model with observed spectra and non-ALA measured traits. They then tested the feasibility of the method using ground spectra and measured CCC data. Nevertheless, there has been no investigation into whether this method can be applied to LAI estimation. Additionally, the strategy of using the adjusted ALA (ALA$_{adj}$) for canopy parameter estimation requires further testing of crop types, sample sizes, and machine learning algorithms.

The purpose of this study is to improve the accuracy of retrieving CCC and LAI for different crops using machine learning algorithms. To achieve this, we proposed the use of adjusted ALA to generate simulations for wheat, soybean, and maize using the PROSAIL model. Subsequently, five machine learning algorithms, namely SVR, RFR, ETR, GBRT, and STL, were used to develop CCC and LAI retrieval models based on the PROSAIL simulation data. Finally, ground measurements were employed to assess the accuracy of the retrieval models.

## 2. Materials and Methods

### 2.1. Study Sites

Datasets of ground measurements for wheat, maize, and soybean crops from experiments conducted in Beijing, China, and Nebraska, United States, were used in this study to validate the retrieval algorithms. The site of the wheat crop is located at the National Station for Precision Agriculture, Xiaotangshan (XTS), Beijing, China (40°10′48″N, 116°26′24″E). In the 2002 campaign, 48 plots of wheat were cultivated with four nitrogen fertilization densities and four water treatment schemes, while, in the 2004 campaign, 42 plots of wheat were planted with non-differentiated fertilization and irrigation management [57]. Three AmeriFlux sites (US-Ne1, US-Ne2, and US-Ne3) at the University of Nebraska–Lincoln Agricultural Research and Development Center near Mead, Nebraska, United States, comprised the maize and soybean study fields. US-Ne1 was irrigated with a center-pivot system while being planted with continuous maize cropping, and field experiments were conducted from 2001 to 2005. In US-Ne2 and US-Ne3, there was a rotation of maize and soybeans, with maize being planted in odd-numbered years (2003 and 2005) and soybean being planted in even-numbered years (2002 and 2004). While US-Ne3 relied only on rainfall for moisture, the same strategy of irrigation as that for US-Ne1 was followed for US-Ne2 [58]. Table 1 provides detailed information for the four sites.

**Table 1.** Details of the study sites.

| Site Name | Country | Latitude/Longitude (°) | Crop Species | Landcover (ha) | Sampling Periods |
|---|---|---|---|---|---|
| XTS | China | 40.18/116.44 | Wheat | 167 | 4–5/2002; 4–5/2004 |
| US-Ne1 | America | 41.17/−96.48 | Maize | 48.7 | (6–9)/(2001–2005) |
| US-Ne2 | America | 41.165/−96.47 | Soybean and maize | 52.4 | 6–9/2002; 6–9/2004 |
| US-NE3 | America | 41.18/−96.44 | Soybean and maize | 65.4 | 6–9/2002 |

### 2.2. Canopy Reflectance Measurements

An ASD FieldSpec Pro spectrometer (Analytical Spectral Devices, Boulder, CO, USA) was implemented to measure the canopy reflectance at the XTS site. The spectrometer measures wavelengths from 350 to 2500 nm and has spectral resolutions of 3 nm between 350 and 1050 nm and 10 nm between 1050 and 2500 nm. Measurements of the canopy spectral properties were taken between 10:00 and 14:00 local time under clear and cloudless conditions. The canopy spectra of wheat were acquired at 1.3 m above the canopy with a field of view of 25 degrees [59]. Two inter-calibrated Ocean Optics USB2000 radiometers (Dunedin Ocean Optics, Dunedin, FL, USA) were used to measure the canopy reflectance spectra at the US-Ne1, US-Ne2, and US-Ne3 sites [60]. The radiometers have a spectral resolution of 1.5 nm and range from 400 to 1100 nm. For one radiometer, upwelling radiation was measured using an optical fiber pointed upward, while downwelling radiation was measured using an optical fiber and a cosine diffuser. The canopy reflectance was then computed using the simultaneous measurements of upwelling radiance and downwelling irradiance.

### 2.3. Measurements of the CCC and LAI

At the XTS site, wheat leaves were collected from the top of the canopy on each plot and were then placed quickly in an ice-filled plastic box for transportation to the laboratory for the measurement of crop parameters. Spectrophotometer measurements were taken of the sample leaves to determine the chlorophyll concentration [61]. Dried-weight measurements were used to determine the green LAI [62]. At the US-Ne2 and US-Ne3 sites, the LCCs of fresh soybean and maize leaves were determined using a spectrophotometer [61]. Using an area meter (Model LI-3100, Li-Cor, Inc., Lincoln, NE, USA), the destructive determination of the green LAI was carried out in the lab (see the research by Viña et al. [63] for more details).

### 2.4. Spectra Simulation Datasets

The canopy spectral characteristics were simulated by the PROSAIL-D model, which was characterized by the coupling of the PROSPECT-D leaf model [64] and the 4SAIL canopy model [65]. Based on the input parameters reported in Table 2, crop canopy spectra in the wavelength range of 400–2500 nm were obtained to train the crop parameter inversion model. The PROSPECT-D model simulated the leaf spectral properties with the set leaf parameters, which were used as an input for the SAIL model.

**Table 2.** Parameter configuration for the PROSAIL-D model.

| | Parameters | Description | Units | Range |
|---|---|---|---|---|
| Leaf | N | Leaf structure index | - | 1, 1.5, 2 |
| | LCC | Leaf chlorophyll content | $\mu g\ cm^{-2}$ | 10~80; interval, 10 |
| | $C_m$ | Leaf dry matter content | $g\ cm^{-2}$ | 0.003, 0.004, 0.005, 0.006 |
| | $C_b$ | Leaf brown pigment content | - | 0 |
| | $C_w$ | Equivalent water thickness | cm | 0.02 |
| | Car | Leaf carotenoid content | $\mu g\ cm^{-2}$ | 25% LCC |
| | $C_{Ant}$ | Leaf anthocyanin content | $\mu g\ cm^{-2}$ | 2 |
| Canopy | LAI | Leaf area index | $m^2\ m^{-2}$ | 0.5, 1, 2, 3, 4, 5, 6, 7, 8 |
| | $\alpha_{soil}$ | Soil reflectance | - | Five soil reflectance types |
| | ALA | Average leaf angle | Degrees | 10–80 degrees |
| | hotS | Hot spot parameter | $m\ m^{-1}$ | 0.05 |
| | skyl | Fraction of diffuse incoming solar radiation | - | 0.5 |
| Observed Geometry | θs | Solar zenith angle | Degrees | 0, 10, 20, 30, 40, 50, 60 |
| | θv | View zenith angle | Degrees | 0 |
| | φ | Sun-sensor azimuth angle | Degrees | 0 |

At the leaf level, the LCC values were set to between 10 and 80 $\mu g\ cm^{-2}$, while carotenoids accounted for 25% of the LCC and varied with the LCC. The leaf structure parameters (N) were set to between 1.0 and 2.0 to represent different leaf thicknesses. The leaf dry matter content ($C_m$) was set to between 0.003 and 0.006 $g\ cm^{-2}$. The equivalent water thickness, leaf anthocyanin content, and leaf brown pigment were set to fixed values, as neither the retrieval CCC nor the LAI is affected by the representation of these parameters on the canopy spectrum.

At the canopy level, there were nine levels of vegetation coverage represented by LAI values ranging from 0.5 to 8. The fraction of diffuse incoming solar radiation (skyl) was fixed at a value of 0.5. Additionally, there were six different ALAs and five different forms of soil reflectance to depict various canopy structures and soil backgrounds.

The ALA was determined by an LAD function, which was represented by LIDFa and LIDFb in the SAIL model. Six types of leaf inclination angle characteristics are commonly used in the SAIL model, and the average value of each type is as follows: planophile, 26.76 degrees; erectophile, 63.24 degrees; extremophile, 45 degrees; plagiophile, 45 degrees; uniform, 45 degrees; spherical, 57.3 degrees. Five soil reflectance types were determined using the measured spectra and five gradients of soil factors (0.1, 0.25, 0.5, 0.75, and 1) as the background parameters of the PROSAIL-D model [66]. Moreover, the observation zenith angle was set as 0 degrees, and the variable solar zenith angle was set as 0–60 degrees with 10 intervals.

### 2.5. Leaf Inclination Angle Optimization Datasets

It is essential to recognize that the canopy structure has a big impact on how effectively crop parameter retrieval works. Although the SAIL model takes six different ALAs into

account to represent the spatial distribution of crop leaves, the actual non-uniform structure of the crop does not correspond to the assumptions of the model. In the PROSAIL model, the ALA is determined by two functions, namely LIDFa and LIDFb, which allows the input of the ALA to be reasonably adjusted to achieve a simulated spectrum that is closer to the actual crop spectrum. Because LIDFb has very little influence on canopy reflectance [30], the ALA has primarily been determined by the LIDFa as ALA = $45 - 360 \times$ LIDFa/$\pi^2$ [67]. According to Jiao et al. [19], the ALA reported in Table 1 was adjusted (ALA$_{adj}$) using the RF algorithm based on the measured data. As a result, ALA$_{adj}$ can be expressed as follows:

$$ALA_{adj} = RFR(SPEC, LAI_m, LCC_m), \tag{1}$$

where *SPEC* is the canopy reflectance measured in the field, and $LAI_m$ and $LCC_m$ are the ground-based LAI and LCC measurements, respectively. The PROSAIL model was updated to input ALA$_{adj}$, which generated simulated canopy spectra adjusted for each crop.

### 2.6. Modeling and Validating Method

Canopy spectral simulation datasets of wheat, soybean, and maize were developed from the PROSAIL model by adjusting the leaf inclination angle. Four machine learning algorithms were used to construct crop parameter retrieval models, and the learning result was integrated using a stacking algorithm. Finally, a total of five retrieval models were constructed. To investigate whether adjusting the ALA enhances the performance of these models, the dataset generated using the parameters presented in Table 1 was used as an alternate model for comparison. A flowchart of the research method can be seen in Figure 1.

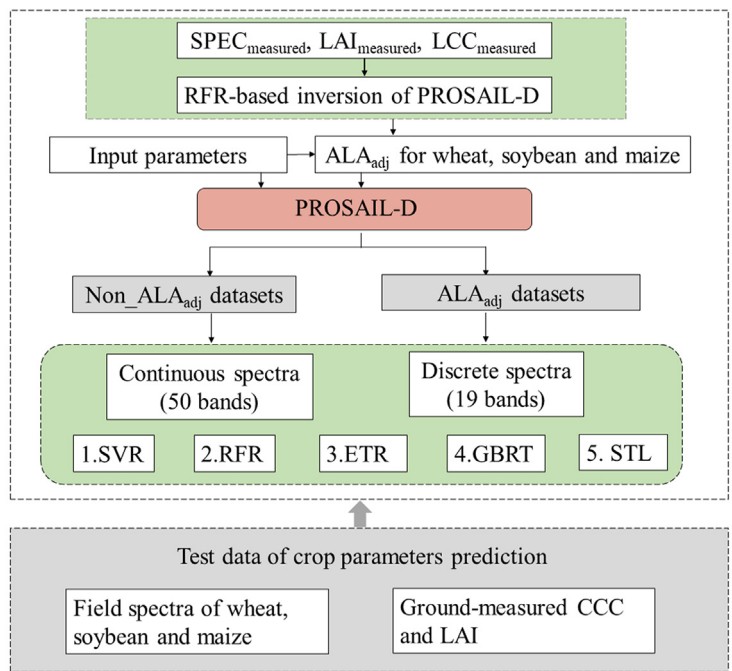

**Figure 1.** Flowchart of the methods to retrieve and validate crop parameters.

### 2.6.1. Machine Learning Retrieval Model

Support vector regression (SVR) is a regression version of the SVM based on statistical learning theory [68]. This algorithm creates linear dependencies between *n*-dimensional input variables and one-dimensional target variables by fitting an adjusted hyperplane to those features. An SVR kernel can be linear, polynomial, Gaussian, or sigmoid [69,70]. By considering the characteristics of the data and comparing the performance of the four kernels, the linear kernel was selected for use in this study. The RFR method is an integrated learning method based on multiple decision trees that combines Breiman's bagging theory

with the random selection of features [38]. To construct the RFR model, it is necessary to determine how many decision trees and variables are required in the bagging framework, and these two values were set to 100 and 10 in this study. ETR is a variant of RFR. RFR uses bootstrap random sampling to select the training set for each decision tree, while ETR does not, which means that each tree is trained from the same dataset. A GBRT algorithm consists of multiple decision trees that are iteratively processed. GBRT generates the entire forest by generating decision subtrees one by one, and the process of generating new subtrees is to use the residuals between the sample label values and the predicted values of the current forest. Essentially, training a model involves the optimization of an objective function that can be derived in any way.

Stacking is a multi-layer model in which several models that have been trained are used as the base model. The prediction results of these several base models are then used as a new training set to train a new model. Most of the time, the models in the first stacking layer are those that have a good fit (such as the SVR, RFR, ETR, and GBRT models used in this study) to pursue the sufficient learning of the training data. Because different models differ in principle, the first-layer model can be considered as the process of automatically extracting effective features from the original data. For the first-layer model, 50 consecutive bands were used as feature variables, and these data were subdivided into 10 folds. For the SVR, RFR, ETR, and GBRT models, 10 training runs were respectively conducted. One-tenth of the samples of each training run were reserved for testing during training, and the prediction of the testing data was performed after the training was completed. Four models were averaged after 10 runs, and their predictive results were stitched together and applied to the training data set in subsequent runs. In the first-layer model, stacking is more prone to overfitting due to the use of complex nonlinear variations for feature extraction. Typically, the second-layer model is a simple one, such as the linear regression model used in this study, to reduce overfitting risks.

In this study, simulated datasets with adjusted ALA values were used to retrieve the LAI and CCC values of wheat, soybean, and maize. The training features were 50 continuous spectral bands from 400 to 890 nm at 10 nm intervals. The SVR, RFR, ETR, and GBRT algorithms and a stacking algorithm integrating the models were selected for the implementation. For comparison, the same method was also carried out for a simulation dataset without adjusted ALA.

### 2.6.2. Performance Assessment

CCC and LAI retrieval was performed for wheat, soybean, and maize using ground reflectance data, and the accuracies were validated using ground-measured data. The performance of the results was assessed using statistical metrics including the coefficient of determination ($R^2$; Equation (2)), root mean square error (RMSE; Equation (3)), bias (Bias; Equation (4)), and normalized RMSE (NRMSE; Equation (5)).

$$R^2 = \frac{\sum_i^n (\hat{y}_i - \overline{y})}{\sum_i^n (y_i - \overline{y})} \tag{2}$$

$$RMSE = \sqrt{\frac{\sum_{i=1}^n (\hat{y}_i - y_i)^2}{n}} \tag{3}$$

$$Bias = \frac{\sum_i^n (y_i - \hat{y}_i)}{n} \tag{4}$$

$$NRMSE = \frac{RMSE}{y_{max} - y_{min}} \tag{5}$$

In these equations, $\hat{y}_i$ represents the predicted CCC value, $y_i$ represents the measured CCC value, $y$ represents the mean of the estimated CCC values, and $n$ represents the number of test samples.

## 3. Results

### 3.1. The Sensitivity of Canopy Reflectance Spectra to Crop Parameters

Figure 2 illustrates the sensitivity of multiple crop parameters to visible and near-infrared (NIR) reflectance. Similar patterns were found in the sensitivities of the CCC, LCC, $C_m$, N, and LAI to the 50 bands for wheat, soybean, and maize with the $ALA_{adj}$. Across the three datasets, the CCC and LAI exhibited strong correlations in the NIR and red-edge regions, whereas the LCC, $C_m$, and N exhibited weak correlations. The variations in the CCC, LCC, and LAI were found to have large effects on the full-band reflectance, while the variations in $C_m$ and *N* have lesser effects on all visible and NIR bands. Regarding the CCC and LAI of concern in this study, the sensitivities of the CCC and LAI of wheat and soybean with the $ALA_{adj}$ to the full-band reflectance were better than those of maize. To achieve the uniformity of the training data of the three crops, the 50 bands were finally selected. It should be noted that the correlation between the LAI and NIR was significantly better than that in the visible region; however, some bands in the visible region remained subject to LAI variation and could thus be used as characteristic bands for the retrieval of the LAI.

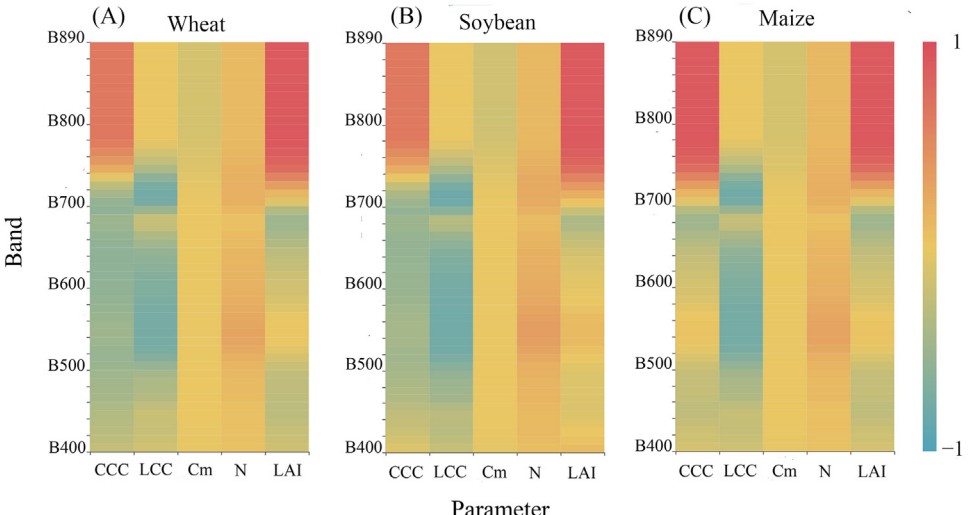

**Figure 2.** The sensitivity analysis for canopy reflectance at various wavelengths in relation to crop parameters from different datasets: (**A**) for wheat; (**B**) for soybean; (**C**) for maize. The color bar represents the magnitude of the correlation between crop parameters and reflectance.

### 3.2. $ALA_{adj}$ Parameterization of Wheat, Soybean, and Maize from PROSAIL Models

Because the vegetation LAD assumed by the PROSAIL model is not consistent with the actual situation, the simulated spectral data and the measured LAI and LCC were used to build an ALA optimization model using the RFR algorithm (Section 2.5). To investigate the role of the measured LAI and LCC in the ALA optimization model, two different types of training data were used in this study. The first dataset contained simulated spectra as well as measured LAI data, while the second incorporated measured LCC data. Both types of data were used in PROSAIL optimization, resulting in very similar wheat $ALA_{adj}$ values (Table 3). A priori knowledge of the leaf inclination angle of wheat led to 62 degrees being used as the $ALA_{adj}$ in this study. As a result of the two datasets, the obtained soybean $ALA_{adj}$ values were respectively 45.1 and 45.7 degrees. In this study, 45 degrees was considered to be the a priori knowledge for the soybean $ALA_{adj}$. The two datasets produced maize $ALA_{adj}$ values of 60.32 and 59.83 degrees, respectively, and 60 degrees was used as the a priori knowledge of the maize $ALA_{adj}$ in this study. For wheat, soybean, and maize, the $ALA_{adj}$ results were stable regardless of whether the LCC was involved. It is noteworthy that the $ALA_{adj}$ values of maize and wheat were relatively close to each other and differed more from those of soybean.

**Table 3.** ALA$_{adj}$ (degrees) based on the measured data of wheat, soybean, and maize, as determined via PROSAIL parameterization.

| Crop Type | Using SPEC_LAI | | Using SPEC_LAI_LCC | | This Study |
|---|---|---|---|---|---|
| | **Mean** | **StDv** | **Mean** | **StDv** | |
| Wheat | 61.8 | 5.2 | 62.0 | 5.2 | 62 |
| Soybean | 45.7 | 9.3 | 45.1 | 9.3 | 45 |
| Maize | 59.8 | 8.1 | 60.3 | 8.1 | 60 |

SPEC LAI refers to measured data for ALA optimization, including canopy spectra and the measured LAI; SPEC_LAI_LCC includes canopy spectra, the measured LAI, and the leaf chlorophyll content (LCC); StDv, standard deviation.

### 3.3. Performance of Machine Learning Algorithms for CCC Retrieval

Table 4 reports the performance of five machine learning approaches for the estimation of the CCC values of wheat, soybean, and maize using the ALA$_{adj}$ and non-ALA$_{adj}$ datasets. According to the validation results for the wheat samples, the RFR, ETR, GBRT, and STL algorithms using the ALA$_{adj}$ datasets achieved significantly superior performance as compared to those using the non-ALA$_{adj}$ dataset. The STL algorithm achieved the highest prediction accuracy with an RMSE of 34.8 µg cm$^{-2}$. Similarly, for the soybean samples, these algorithms using the proposed soybean ALA$_{adj}$ datasets produced significantly better validation results than those using the non-ALA$_{adj}$ dataset, and the ETR algorithm had the highest retrieval accuracy. For the maize samples, the SVR, RFR, and STL algorithms using the maize ALA$_{adj}$ datasets outperformed those using the non-ALA$_{adj}$ dataset. Among all the algorithms, the SVR algorithm delivered the best results in the estimation of the maize CCC with the measured CCC values; it reached 433.3 µg cm$^{-2}$ at the maximum and 215.7 µg cm$^{-2}$ on average. For the wheat validation samples, it was found that the SVR method performed just marginally better when using the non-ALA$_{adj}$ dataset than it did when using the ALA$_{adj}$ dataset. Additionally, the ETR and GBRT methods using the non-ALA$_{adj}$ dataset only slightly outperformed those using the ALA$_{adj}$ dataset in the case of the maize samples. Conversely, RFR, which is comparable to the ERT and GBRT algorithms, performed substantially better with the ALA$_{adj}$ datasets than without them. There was a wide variation in the CCC prediction results of the five machine learning algorithms for the three crops, but the retrieval results of the RFR algorithm for wheat and soybean were similar to those found by Jiao [19]. The ETR, GBRT, and STL algorithms produced better validation results than the RFR algorithm, which improved the retrieval accuracy for soybean and wheat. Although some individual models achieved better performance with non-adjusted data, across the three crops, all five machine learning algorithms performed more consistently and stably with the ALA$_{adj}$ datasets.

The validation plots of the CCC retrieval based on continuous ground spectra for mixed crops are presented in Figure 3. Minimal differences were found between adjusting and not adjusting the ALA for SVR prediction, as shown in Figure 3A,F. Figure 3B,G show that the CCC retrieval accuracy of the RFR algorithm was considerably increased after ALA optimization, giving a decreased CCC RMSE value of 51.1 µg cm$^{-2}$ against 93.0 µg cm$^{-2}$. For the RFR algorithm, the non-ALA$_{adj}$ dataset resulted in greater overestimation and dispersion for all three crops, whereas the ALA$_{adj}$ dataset resulted in a more effective correction for overestimation. On the basis of the comparison of Figure 3C,H, it is evident that the ETR predicted the CCC for mixed crops more accurately with the ALA$_{adj}$ approach. As presented in Figure 3D,I, the GBRT and ERT methods exhibited similar patterns and predictability. As a result of the integration of the SVR, RFR, ETR, and GBRT, the STL model was subject to the performance of these machine learning algorithms. The comparison of Figure 3E,F reveals that the adjusted ALA significantly improved the retrieval accuracy of the mixed crops based on the STL algorithm (ALA$_{adj}$: $R^2$ = 0.74, RMSE = 48.3 µg cm$^{-2}$, Bias = 3.6 µg cm$^{-2}$, NRMSE = 9%; non-ALA$_{adj}$: $R^2$ = 0.63, RMSE = 68.5 µg cm$^{-2}$, Bias = $-30.8$ µg cm$^{-2}$, NRMSE = 16%).

**Table 4.** Statistical results of CCC retrieval using ALA$_{adj}$ and comprehensive dataset for wheat, soybean, and maize.

| ML Algorithm | Crop Type | ALA$_{adj}$ Dadaset | | | | Non-ALA$_{adj}$ Dataset | | | |
|---|---|---|---|---|---|---|---|---|---|
| | | $R^2$ | RMSE | Bias | NRMSE | $R^2$ | RMSE | Bias | NRMSE |
| SVR | Wheat | 0.59 | 41.2 | −25.5 | 16% | 0.62 | 31.7 | −6.6 | 13% |
| | Soybean | 0.84 | 46.1 | −29.3 | 17% | 0.85 | 49.5 | −28.2 | 18% |
| | Maize | 0.86 | 49.3 | 23.4 | 12% | 0.86 | 50.8 | 27.6 | 12% |
| RFR | Wheat | 0.54 | 39.4 | −17.4 | 16% | 0.2 | 97.2 | −57.7 | 38% |
| | Soybean | 0.84 | 50.8 | −25.8 | 19% | 0.75 | 72.4 | −43.8 | 26% |
| | Maize | 0.78 | 64.3 | 19.5 | 15% | 0.53 | 93.8 | −10.4 | 22% |
| ETR | Wheat | 0.48 | 37.4 | 7.1 | 15% | 0.37 | 48.0 | −4.7 | 19% |
| | Soybean | 0.83 | 34.0 | −9.3 | 13% | 0.85 | 72.2 | −44.8 | 27% |
| | Maize | 0.73 | 77.6 | 43.1 | 18% | 0.68 | 69.7 | 18.2 | 16% |
| GBRT | Wheat | 0.48 | 37.4 | 7.0 | 14% | 0.37 | 48.0 | −4.7 | 19% |
| | Soybean | 0.83 | 34.0 | −9.3 | 13% | 0.86 | 72.2 | −44.8 | 26% |
| | Maize | 0.74 | 77.6 | 43.1 | 19% | 0.68 | 69.7 | 18.2 | 17% |
| STL | Wheat | 0.58 | 34.8 | −7.7 | 14% | 0.38 | 63.8 | −39.5 | 25% |
| | Soybean | 0.84 | 45.7 | −21.7 | 17% | 0.85 | 55.3 | −35.9 | 20% |
| | Maize | 0.80 | 63.4 | 29.4 | 15% | 0.63 | 79.0 | −16.3 | 19% |

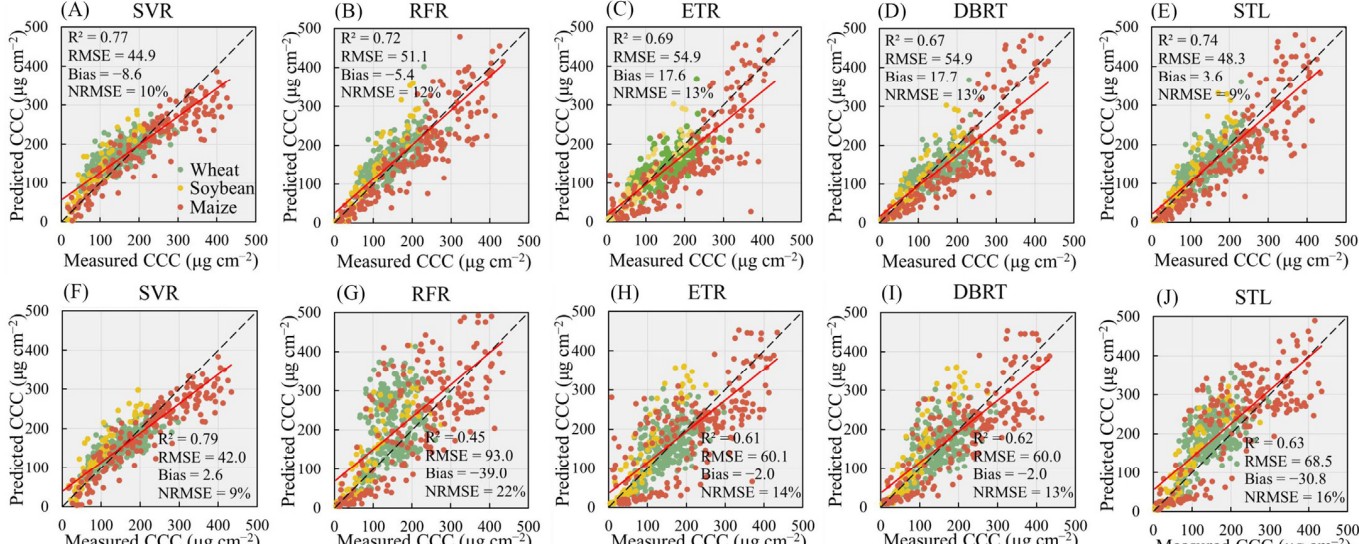

**Figure 3.** The scatter plots of the ground-truth CCC and the predicted CCC from the ground-based spectra. (**A–E**) Performances of SVR, RFR, ETR, GBRT, and STL algorithms based on adjusted ALA dataset, respectively; (**F–J**) performances of SVR, RFR, ETR, GBRT, and STL algorithms based on non-adjusted dataset, respectively.

### 3.4. Performance of Machine Learning Algorithms for LAI Retrieval

In this study, the adjusted ALA method was applied to LAI estimation to investigate its performance in LAI retrieval. According to Table 5, five machine learning algorithms were assessed using ALA$_{adj}$ and non-ALA$_{adj}$ datasets for the retrieval of the LAIs of wheat, soybean, and maize. In contrast to the CCC retrieval results, the SVR, RFR, ETR, GBRT, and STL methods predicted more accurate LAI values with the adjusted ALA than without it. Using the wheat ALA$_{adj}$ training data given in this study, the SVR, RFR, ETR, GBRT, and

STL algorithms exhibited similar retrieval accuracies for wheat samples with RMSE values ranging from 0.6 to 0.8, and the SVR algorithm achieved the highest validation accuracy (ALA$_{adj}$: $R^2$ = 0.46, RMSE = 0.62, Bias = 0.19, NRMSE = 15%; non-ALA$_{adj}$: $R^2$ = 0.47, RMSE = 0.99, Bias = 0.64, NRMSE = 24%). For the soybean LAI, the SVR, RFR, ETR, GBRT, and STL algorithms using the adjusted ALA achieved closer validation results, with RMSE distributions around 0.80 (RMSE: SVR = 0.87, RFR = 0.83, ETR = 0.86, GBRT = 0.87, STL = 0.78). In the case of the non-adjusted ALA, the validation results for the soybean LAI were poor, and the RMSE values of all five machine learning algorithms exceeded 1 (RMSE: SVR = 2.73, RFR = 1.80, ETR = 1.67, GBRT = 1.68, STL = 1.35). Similar prediction accuracies were achieved by the SVR, RFR, ETR, GBRT, and STL algorithms using the adjusted ALA for maize, with RMSE values ranging from 1.1 to 1.4. In contrast, none of the five algorithms performed well when tested on the non-adjusted ALA. The validated RMSE values for maize were greater than those for wheat and soybean, which was attributed to the larger LAI of the maize sample. Using the adjusted ALA, the five machine learning algorithms predicted the LAI of the three crops with less variation and better performance than when using the non-adjusted ALA, indicating that the adjusted ALA is effective and stable for the LAI estimation of crops.

**Table 5.** Statistical results of LAI retrieval using ALA$_{adj}$ and comprehensive dataset for wheat, soybean, and maize.

| ML Algorithm | Crop Type | ALA$_{adj}$ Dadaset | | | | Non-ALA$_{adj}$ Dataset | | | |
|---|---|---|---|---|---|---|---|---|---|
| | | $R^2$ | RMSE | Bias | NRMSE | $R^2$ | RMSE | Bias | NRMSE |
| SVR | Wheat | 0.46 | 0.62 | 0.19 | 15% | 0.47 | 0.99 | 0.64 | 24% |
| | Soybean | 0.87 | 0.87 | 0.53 | 16% | 0.85 | 2.73 | 1.00 | 48% |
| | Maize | 0.88 | 1.14 | 0.84 | 21% | 0.55 | 2.84 | 1.77 | 46% |
| RFR | Wheat | 0.43 | 0.69 | 0.34 | 17% | 0.13 | 2.45 | −1.49 | 60% |
| | Soybean | 0.75 | 0.83 | 0.43 | 16% | 0.54 | 1.80 | −0.84 | 34% |
| | Maize | 0.70 | 1.33 | 1.02 | 25% | 0.29 | 2.18 | −0.36 | 37% |
| ETR | Wheat | 0.47 | 0.65 | 0.31 | 16% | 0.27 | 1.66 | −1.20 | 41% |
| | Soybean | 0.75 | 0.86 | 0.51 | 16% | 0.77 | 1.67 | −1.03 | 31% |
| | Maize | 0.71 | 1.40 | 1.12 | 26% | 0.37 | 1.67 | 0.01 | 28% |
| GBRT | Wheat | 0.48 | 0.66 | 0.30 | 15% | 0.27 | 1.67 | −1.21 | 41% |
| | Soybean | 0.75 | 0.87 | 0.50 | 16% | 0.77 | 1.68 | −1.02 | 32% |
| | Maize | 0.70 | 1.40 | 1.13 | 16% | 0.38 | 1.67 | 0.01 | 28% |
| STL | Wheat | 0.51 | 0.64 | 0.33 | 16% | 0.27 | 1.31 | −0.79 | 32% |
| | Soybean | 0.79 | 0.78 | 0.43 | 15% | 0.68 | 1.45 | −0.68 | 27% |
| | Maize | 0.74 | 1.26 | 0.97 | 23% | 0.28 | 1.78 | 0.36 | 30% |

Figure 4 displays the validation plots for the LAI retrieval using continuous ground spectra for mixed crops. A comparison of the validation scatters obtained using the ALA$_{adj}$ and non-ALA$_{adj}$ datasets shown in Figure 4 indicates that the use of the adjusted ALA made the predicted LAI values more consistent with the measured LAI values. Figure 4A,F reveal a significant improvement in the prediction accuracy of the SVR algorithm for the LAI of mixed crops when using ALA optimization, with the RMSE values decreased from 2.10 to 0.91. The improvement of the adjusted ALA for the RFR, ETR, GBRT, and STL algorithms was better than that of the non-adjusted ALA, especially for soybean and wheat. Although the adjusted ALA improved the accuracy of maize LAI retrieval by the RFR, ETR, GBRT, and STL algorithms, underestimation remained. Based on the combination of the validation results of the five algorithms, SVR performed better than the other algorithms in retrieving the mixed-crop LAIs (RMSE: SVR = 0.91, RFR = 1.03, ETR = 1.05, GBRT = 1.05, STL = 0.97).

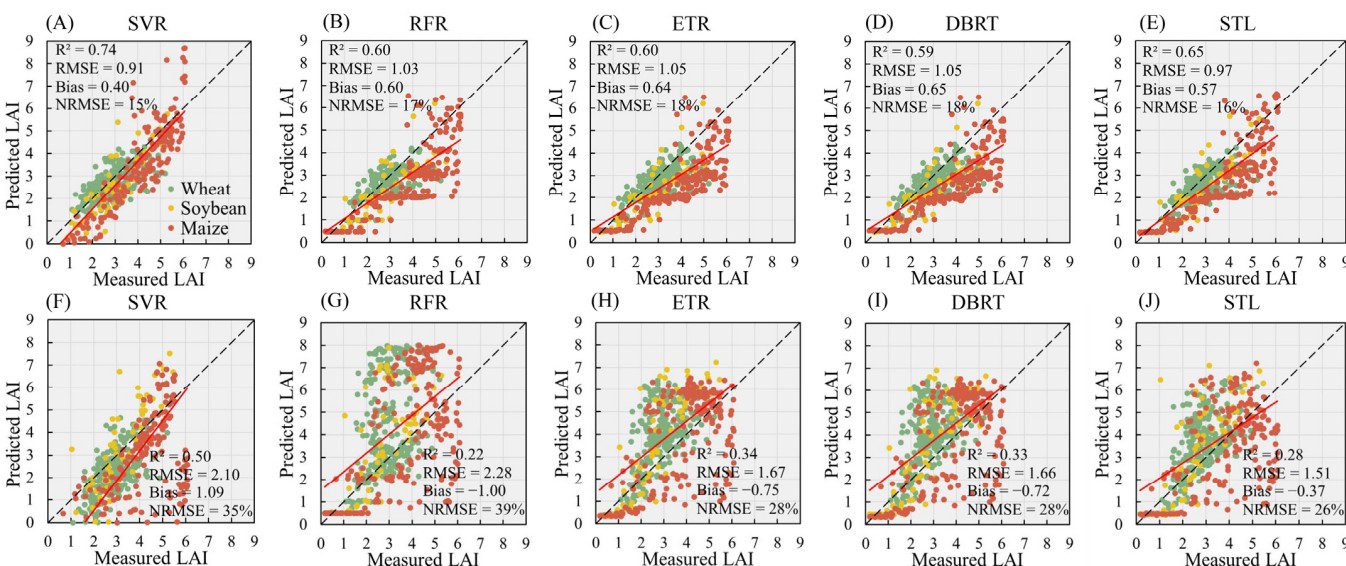

**Figure 4.** Scatter plots of ground-truth LAI and LAI predicted from ground-based spectra. (**A–E**) Performances of SVR, RFR, ETR, GBRT, and STL algorithms based on adjusted ALA dataset, respectively; (**F–J**) performances of SVR, RFR, ETR, GBRT, and STL algorithms based on non-adjusted dataset, respectively.

## 4. Discussion

### 4.1. Application of the $ALA_{adj}$ Parameterization of the PROSAIL Model to Crop CCC and LAI Retrieval

In this study, an $ALA_{adj}$ parameterization scheme was used to improve the performance of the PROSAIL model in retrieving the CCC and LAI of crops. The PROSAIL model assumes a uniform canopy, thus limiting its application in non-uniform canopy layers. As a structural parameter of the PROSAIL model, the ALA describes the random spatial distribution of leaves. The ALA setting determines how well the spectrum is transmitted within the vegetation, thus affecting its range of reception [71]. In Sections 3.3 and 3.4, it was demonstrated that the retrieval accuracy of the CCC and LAI can be improved by optimizing the ALA for different vegetation types. In a previous study, Jiao et al. [19] provided the CCC estimates for wheat based on an adjusted ALA using the RF algorithm (RMSE = 37.9 µg cm$^{-2}$), which is similar to the retrieval results of the RFM in the present study (RMSE = 39.42 µg cm$^{-2}$). It is important to note, however, that the ETR, GBRT, and STL algorithms used in this study produced better estimates of the wheat CCC than Jiao et al.'s [11] model (RMSE: ETR = 37.41 µg cm$^{-2}$, GBRT = 37.42 µg cm$^{-2}$, STL = 34.81 µg cm$^{-2}$). Although the soybean estimates obtained by Jiao et al. [19] using the RF model (RMSE = 39.9 µg cm$^{-2}$) were more accurate than those produced by the RFR algorithm in this study (RMSE = 50.81 µg cm$^{-2}$), the ETR and GBRT algorithms provided superior estimates (RMSE: ETR = 33.98 µg cm$^{-2}$; GBRT = 33.99 µg cm$^{-2}$). It appears that optimizing the ALA can improve the accuracy of crop CCC estimation, although different machine learning algorithms will achieve different performances.

In this study, for the first time, LAI estimation using ALA parameterization was successfully completed. The adjusted ALA method has never been used for LAI estimation, but the findings of this study indicate that the accuracy of the five machine learning algorithms increased significantly after ALA optimization. For example, for the RFR method, which is commonly used for quantitative vegetation retrieval, the parameterization of the ALA for the three crops produced significantly better LAI estimates than the results without optimization ($ALA_{adj}$: RMSE$_{Wheat}$ = 0.69, RMSE$_{Soybean}$ = 0.83, RMSE$_{Maize}$ = 1.33; non-$ALA_{adj}$: RMSE$_{Wheat}$ = 2.45, RMSE$_{Soybean}$ = 1.80, RMSE$_{Maize}$ = 2.08). In addition, the rest of the algorithms also achieved effective and stable retrieval performance. Compared to CCC estimation, ALA parameterization contributes more to LAI retrieval, resulting in

the more stable prediction performance of multiple machine learning algorithms. Figure 5 presents the prediction accuracy of the five machine learning algorithms for wheat, soybean, and maize LAI estimation using adjusted and non-adjusted ALA datasets, with the RMSE as the evaluation criterion. It is evident from the figure that for all three crop types, the accuracy of LAI estimation using the adjusted ALA method was significantly higher than that using the non-adjusted ALA method. Moreover, the prediction accuracies of the five machine learning algorithms for the same crops with the parametric ALA did not differ significantly, whereas the non-adjusted ALA methods exhibited a large difference. The boxplot shown in Figure 6 reveals the distribution of the differences between the measured and predicted LAI values. It is evident from the figure that the differences between the measured and predicted LAI values for wheat and soybean adjusted based on the adjusted ALA were significantly more concentrated, with the maximum and minimum values of the differences being smaller and the median values distributed around 0. In contrast, the differences between the measured and predicted LAI values for wheat and soybean based on the non-adjusted ALA were more discrete and exhibited large outliers. In other words, the predicted LAI values for wheat and soybean without the adjusted ALA presented more overestimations or underestimations. The difference between the measured and predicted LAI values of maize based on the adjusted ALA had a median value greater than 0 across multiple machine learning algorithms, revealing a general underestimation. The difference between the measured and predicted LAI values for maize without the adjusted ALA appeared to have a concentrated distribution. This was due to a larger number of outliers canceling each other out, as explained in Figure 4.

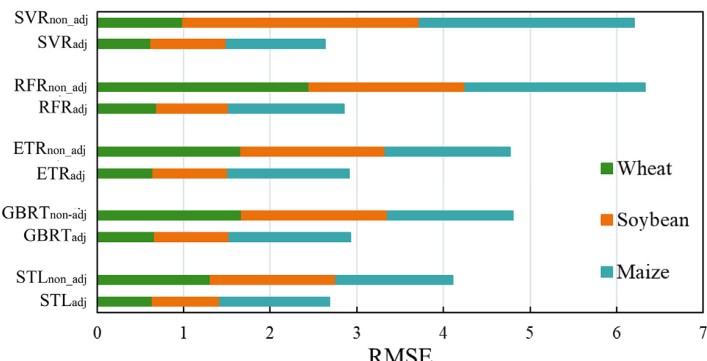

**Figure 5.** Comparison of RMSE for wheat, soybean, and maize using SVR, RFR, ETR, GBRT, and STL algorithms based on adjusted ALA and non-adjusted ALA datasets.

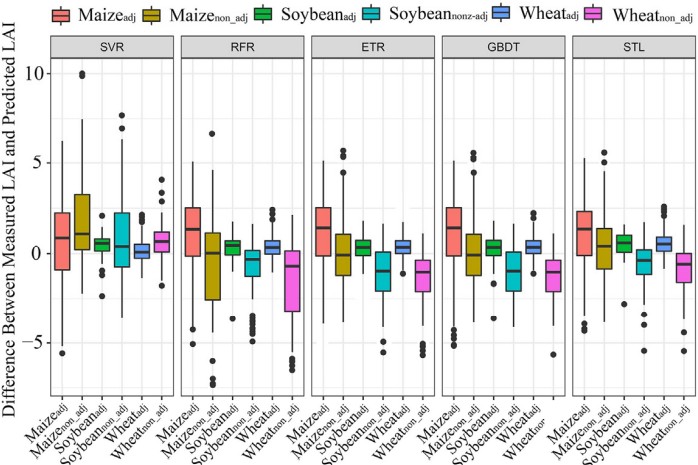

**Figure 6.** Boxplot of the difference between measured LAI and predicted LAI using SVR, RFR, ETR, GBRT, and STL algorithms based on adjusted ALA and non-adjusted ALA datasets.

### 4.2. Performance of the Machine Learning Regression Algorithms for CCC and LAI Retrieval

In this paper, machine learning methods were used to implement the retrieval algorithm for estimating crop CCC and LAI. However, as empirical regression is a convenient method for estimating these parameters, we also compared the performance of commonly used vegetation indices with machine learning algorithms. To estimate the CCC, we used the MTCI, which was developed using red and red-edge position bands. To estimate the LAI, we used the NDVI, which is calculated by analyzing the difference between the reflectance of near-infrared and red-light wavelengths. Empirical models for MTCI-CCC and NDVI-LAI were constructed using the adjusted simulation datasets, and the validation results are presented in Figure 7. The validation results revealed that the empirical model constructed using MTCI produced a significant overestimation of CCC for all three crops, as shown in Figure 7A. A comparison with Figure 3 illustrates that the machine learning algorithms estimated the crop CCC significantly more effectively than the empirical model. Moreover, the regression model constructed using NDVI exhibited a large number of negative values when estimating LAI, and these values are not displayed in Figure 7B. Further comparison with Figure 4 reveals that the performance of the regression model was also weaker than that of the machine learning algorithms.

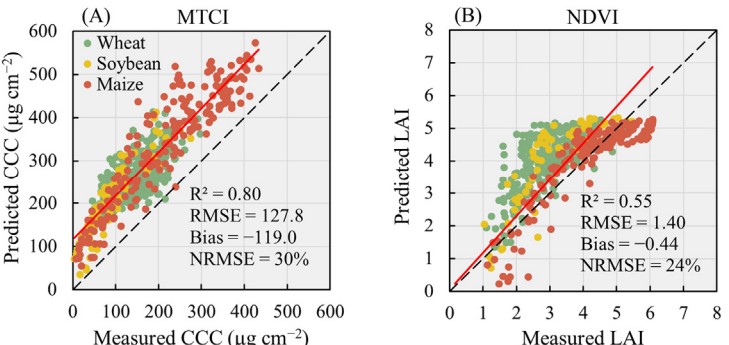

**Figure 7.** The scatter plots for validation results of empirical models based on vegetation indices. (**A**) estimation of CCC performance using MTCI; (**B**) estimation of LAI performance using NDVI.

Machine learning algorithms can efficiently handle complex nonlinear relationships in crop parameter inversions due to their powerful computational capabilities. However, the model outputs may have potential biases and distortions, such as negative LAI estimates, when the created models are given access to real crop parameters; the term for this situation is overfitting [24]. In this work, the SVR algorithm was found to achieve superior CCC and LAI estimates; for the CCC estimation in particular, the adjustment of the ALA had little bearing on the outcomes of the estimation (see Figure 3). However, Figure 3 does not display a negative CCC estimation, which was only evident for the SVR model among the five machine learning algorithms (Figure 8A,B). Figure 8A,B display the scatter plots of the negative predicted values of the SVR algorithm under the adjusted and non-adjusted ALA strategies, respectively. Additionally, on both the adjusted and non-adjusted ALA datasets, the RFR, ETR, GBRT, and STL algorithms displayed differential inversion performance. However, due to the modification of the ALA, particularly for the RFR method, the accuracy of CCC retrieval was significantly improved ($ALA_{adj}$: $R^2 = 0.60$, RMSE = 1.03, Bias = 0.60, NRMSE = 17%; non-$ALA_{adj}$: $R^2 = 0.22$, RMSE = 2.28, Bias = 1.00, NRMSE = 39%). The performances of the SVR, RFR, ETR, GBRT, and STL in estimating crop LAI values differed; the RFR, ETR, GBRT, and STL algorithms exhibited similar inversion results, all of which were weaker than those of the SVR algorithm. However, the LAI calculated by the SVR algorithm displayed negative values, perhaps due to overfitting (Figure 8C,D). Figure 8C,D illustrate that negative values occurred mainly when the measured LAI was small, and also that there were more and larger negative values under the non-adjusted ALA strategy. For wheat and soybean, the RFR, ETR, GBRT, and STL algorithms vastly enhanced the LAI estimations, whereas the maize LAI estimated by employing the modified ALA

strategy remained significantly underestimated (Figure 4). Although the maize LAI was underestimated by the modified ALA strategy, the results were still superior to the discrete LAI estimates obtained under the non-adjusted ALA strategy. The results of this research reveal that crop CCC and LAI estimation differ among the SVR, RFR, ETR, GBRT, and STL regression algorithms and that the method of adjusting the ALA essentially enhances the inversion performance of each algorithm.

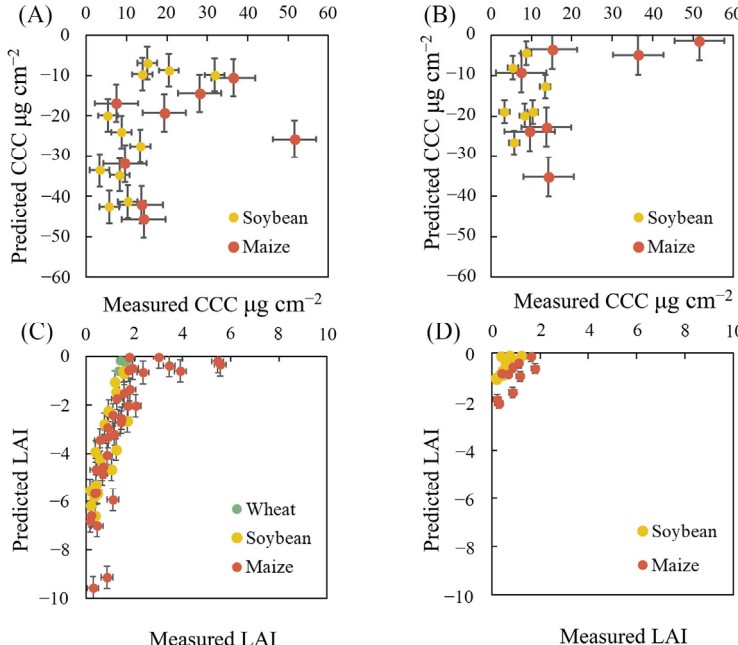

**Figure 8.** Negative predictions of SVR for CCC and LAI: (**A**,**C**) without adjusted ALA strategy; (**B**,**D**) with adjusted ALA strategy.

### 4.3. Effects of the Uncertainties of the Selected Spectra on the Retrieval Models

In this study, 50 continuous spectra were used to estimate the crop canopy parameters by optimizing the ALA. The final results of the retrieval models will be impacted if the training spectra differ. Furthermore, the performances of discrete spectra on the satellite scale were evaluated with an eye toward the future application of retrieval models to satellites. Figure 9 presents the comparison of the validation results of the CCC and LAI retrieval models using continuous and discrete (19 bands) spectra with $ALA_{adj}$ and non-$ALA_{adj}$ datasets. Under the same ALA parameterization conditions, the reflectance of continuous and discrete bands had little effect on the inversion accuracy of the CCC and LAI. Even individual machine learning algorithms exhibited better prediction performance when using discrete reflectance spectra. For example, when using discrete reflectance spectra, the CCC estimations of the RFR, ETR, and GBRT algorithms for wheat, soybean, and maize were slightly better than those when using continuous reflectance spectra. The LAI estimations of the SVR algorithm for wheat, soybean, and maize were slightly better than those when using the continuous spectra. There was, however, no significant difference between the accuracy of the CCC and LAI estimations for the two types of spectra. Nevertheless, the retrieval accuracies of the continuous and discrete bands with and without the adjusted ALA technique were noticeably lower than those of the adjusted ALA technique, with the discrete bands exhibiting a striking difference between the retrieval accuracies of the two strategies. This indicates that the adjusted ALA makes a significant contribution to the discrete band in CCC and LAI estimation. The feasibility of the use of discrete spectra provides selectivity for a variety of training variables. The sensitivity of reflectance to crop parameters and a priori knowledge can lead to better band selection in subsequent studies. Because the surface reflectance of satellite images is characterized by

discrete spectra, the results of this study illustrate that this method may also achieve good results using satellite data.

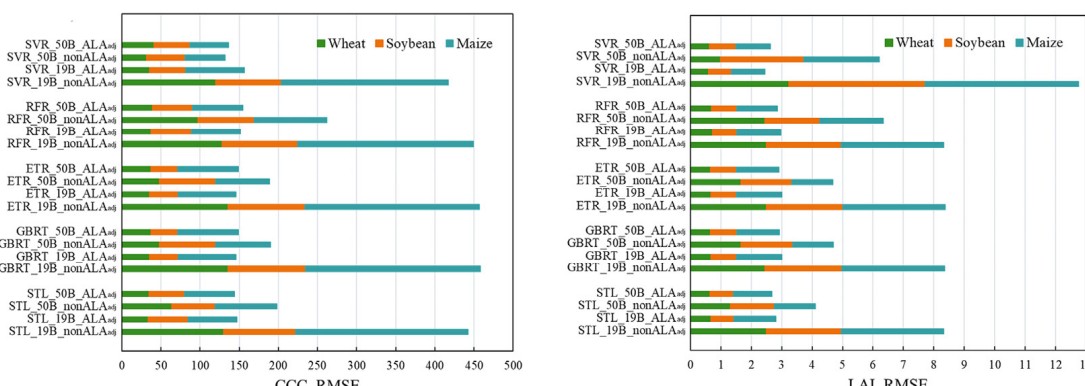

**Figure 9.** Comparison of RMSE of CCC and LAI for wheat, soybean, and maize using continuous and discrete spectra based on SVR, RFR, ETR, GBRT, and STL algorithms.

### 4.4. Challenges and Limitations of the Proposed Retrieval Approach

If the proposed method is applied on the satellite scale, it will be necessary to consider the effect of crop spatial distribution on $ALA_{adj}$ determination. Unlike the present study, which used ground-based spectra to obtain the $ALA_{adj}$, the adjusted ALA faces challenges related to the resolution and spatial heterogeneity of satellite data. Additionally, actual satellite data are subject to a number of uncertainties, including remote sensor hardware metrics, atmospheric correction processing, and complex ground mixing [72,73].

The CCC and LAI retrieval algorithms depend on the parameter inputs of the RTM. Although the retrieval algorithm was adjusted in this study by adjusting the ALA, additional physicochemical parameters of the crops were not explored, resulting in some uncertainty regarding the suggested retrieval algorithms under various crop types and growing circumstances. In areas with low plant cover, the estimation of the canopy parameters has been shown to be a challenge in quantitative remote sensing [74], and the findings of this work are no exception. For instance, when the SVR is used to estimate crop LAI, a significant proportion of negative values will arise in low-LAI locations.

The process of assessing the actual field data could be another source of uncertainty. In this study, three different crop types were used as validation samples, but each crop sample was taken from the same region. The current validation dataset does not accurately reflect the regional and seasonal variability of crops. Future research could concentrate on extending the validation of the retrieval algorithm by utilizing a wider variety of crop samples to assess the accuracy of the CCC and LAI estimation models for multiple crop types.

### 5. Conclusions

This study presented an improved approach for estimating the CCC and LAI of wheat, soybean, and maize by parameterizing the crop-type-related $ALA_{adj}$. The $ALA_{adj}$ values of these crops (62 degrees for wheat, 45 degrees for soybean, and 60 degrees for maize) were obtained using ground-based prior data and a random forest algorithm. The results of five machine learning algorithms, namely SVR, RFR, ETR, GBRT, and STL, were investigated and compared using adjusted and original datasets. The findings indicate that the machine learning algorithms using the $ALA_{adj}$ resulted in a significant accuracy improvement for the estimation of the CCC and LAI of wheat, soybean, and maize, especially the crop LAI. In terms of the estimation results of mixed crops using $ALA_{adj}$ datasets, the SVR algorithm performed the best for both CCC and LAI estimation, with an RMSE of 44.93 μg cm$^{-2}$ for the CCC estimation and an RMSE of 0.91 for the LAI estimation. Additionally, although RFR, ETR, GBRT, and STL slightly underperformed SVR, these

algorithms also have good potential for CCC and LAI retrieval, especially STL, which achieved an RMSE of 48.27 µg cm$^{-2}$ for the CCC estimation and an RMSE of 0.97 for the LAI estimation. Furthermore, the proposed method was applied to 19-band spectra from the reference Sentinel-3 satellite, and similar results were obtained. This indicates the promising potential of the proposed method to estimate the CCC and LAI of crops at the satellite scale. Future studies could explore the feasibility of the proposed method for other crops and regions.

**Author Contributions:** Conceptualization: Q.S., Q.J. and B.Z.; methodology: Q.S. and Q.J.; validation: Q.S. and H.X.; formal analysis: Q.S. and Q.J.; investigation: X.C. and W.H.; data curation: Q.S. and H.X.; writing—original draft preparation: Q.S.; writing—review and editing: X.C.; visualization: Q.S.; supervision: W.H. and B.Z. All authors have read and agreed to the published version of the manuscript.

**Funding:** This research was funded by the National Key Research and Development Program of China, grant number 2021YFB3900501, the National Natural Science Foundation of China, grant numbers 42030111 and 42071330, the Fengyun Application Pioneering Project, grant number FY-APP, and Henan Province Key R&D and Promotion Special Project, grant number 202102110270.

**Data Availability Statement:** All data used in this study were from published field work cited in this article.

**Acknowledgments:** We appreciate the data provided by the Center for Advanced Land Management Information Technologies (CALMIT), University of Nebraska–Lincoln.

**Conflicts of Interest:** The authors declare no conflict of interest.

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
