# Peer review of "Machine Learning Algorithms for the Retrieval of Canopy Chlorophyll Content and Leaf Area Index of Crops Using the PROSAIL-D Model with the Adjusted Average Leaf Angle"

_remotesensing, doi:10.3390/rs15092264_

Round 1
Reviewer 1 Report
The authors used the PROSAIL-D model to generate ALAadj values for wheat, soybean, and corn based on ground measurements of spectra. Then, they estimated crop CCC and LAI using five machine learning approach. This paper needs major revision before it accepted. The following recommendations need to be seriously revised
The authors used the LAI and LCC from ground observations into the PROSAIL model to obtain ALAadj, then used ALAadj inversion to obtain LAI and CCC, and finally considered the retrival accuracy improved. This process is a circular verification, right?
Line 58-76: Please add details about CCC and LAI. In addition, CCC is a key indicator affecting solar-induced chlorophyll fluorescence (SIF) and Vcmax, please refer to the following paper and add to the introduction to increase the novelty and cutting-edge of the paper.
[1] Monitoring drought impacts on crop productivity of the U.S. Midwest with solar-induced fluorescence: GOSIF outperforms GOME-2 SIF and MODIS NDVI, EVI, and NIRv
[2] Leaf chlorophyll content as a proxy for leaf photosynthetic capacity
Line 131-148: Please add an information table about the site, such as site name, latitude, longitude, landcover, and the time of the data. There is no figure about study area throughout the paper, which makes it difficult to read.
Line144-145: US-Ne1,2,3 have flux data from 2001 to 2020, why are only 2001, 2005, 2002, and 2004 used? Are there any special data screening requirements??
Line 154-155: The data from the flux towers are 24-hour observations, why only the midday (from 10:00 to 12:00) data are used here?
Figure 2. Please add wheat, corn, soybean into figure.2 to increase the readability of the graph.
Table3. Why does wheat have two SVR calculations?
Reviewer 2 Report
Dear Respected Authors
The submitted manuscripts lacks novelty/originality and shares too much similar data with your previously published manuscript.
Regards
Reviewer 3 Report
The authors deal with a motivating topic in image retrieval which is a valuable tool in Hyperspectral image processing. I consider the subject significant and valuable revisiting taking into account the new possibilities for Machine Learning Algorithms applications. The paper is acceptable presented, the English are clear, and the overall methodology is well explained and properly referenced. The authors demonstrate a deep knowledge of mathematical methods to retrieve canopy chlorophyll content and leaf area index of crops and clearly describe them. Only minor modifications are needed to clarify some of the concepts and the particular aims of this paper.
Reviewer 4 Report
This manuscript, entitled ‘Machine Learning Algorithms for the Retrieval of Canopy 2 Chlorophyll Content and Leaf Area Index of Crops Using the 3 PROSAIL-D Model with the Adjusted Average Leaf Angle," dealt with crop trait predictions with different machine learning approaches. I found this manuscript interesting and of general interest. But I have a few suggestions that may help the authors improve their manuscript.
First of all, this manuscript is extremely difficult to read due to excessive use of abbreviated forms. Most importantly the abstract sections used numerous short forms without introducing them. I strongly recommend to change this patterns. Abstracts should be understandable to all and stand-alone.
Second, why did the author not apply simple non-linear models before applying ML approaches? It's easier and more convenient for all. So, I would recommend starting the analysis with simple models and then comparing them with ML approaches.
Third, the results section should be improved, specifically the goodness of fit index and units, which sometimes made me confused. Cross-check, please.
Finally, in the conclusion section, authors should have drawn the line that the best parsimonious approach takes. It seems authors repeated their results, so I would recommend rewriting the conclusion parts again, focusing on comparative models.
Round 2
Reviewer 2 Report
Dear Authors
Despite your kind reply, the similarity in both studies is against publication ethics, as I know.
yours
Author Response
Thank you for taking the time to review our manuscript. We appreciate your comments and feedback.
Reviewer 4 Report
Thanks to all authors for answering the comments and your efforts.
Author Response

(The authors gave the same response as above.)
